# The Supported Boro-Additive Effect for the Selective Recovery of Dy Elements from Rare-Earth-Elements-Based Magnets

**DOI:** 10.3390/ma15093032

**Published:** 2022-04-21

**Authors:** Sangmin Park, Dae-Kyeom Kim, Javid Hussain, Myungsuk Song, Taek-Soo Kim

**Affiliations:** 1Research Institute of Advanced Manufacturing Technology, Korea Institute of Industrial Technology, Incheon 21999, Korea; jhsm8920@kitech.re.kr (S.P.); kyeom@kitech.re.kr (D.-K.K.); javidmohsin77@kitech.re.kr (J.H.); 2Industrial Technology, Korea University of Science and Technology, Daejeon 34113, Korea

**Keywords:** pyrometallurgy, recycling, liquid metal extraction, boro-additive effect, Dy–Fe intermetallic compounds

## Abstract

Liquid metal extraction (LME) for recycling rare-earth elements from magnets is studied, in the present study, to examine its suitability as an environmentally friendly alternative for a circular economy. While Nd (neodymium) extraction efficiency can easily reach almost 100%, based on the high reactivity of Mg (magnesium), Dy (dysprosium) extraction has been limited because of the Dy–Fe intermetallic phase as the main extractive bottleneck. In the present paper, the boro-additive effect is designed thermodynamically and examined in the ternary and quinary systems to improve the selectivity of recovery. Based on the strong chemical affinity between B (boron) and Fe, the effect of excess boron, which is produced by the depletion of B in FeB by Mg, successfully resulted in the formation of Fe_2_B instead of Dy–Fe bonding. However, the growth of the Fe_2_B layer, which is the reason for the isolated Mg, leads to the production of other byproducts, rare-earth borides (*R*B_4_, *R* = Nd and Dy), as the side effect. By adjusting the ratio of FeB, the extraction efficiency of Dy over 12 h with FeB addition is improved to 80%, which is almost the same extraction efficiency of the conventional LME process over 24 h.

## 1. Introduction

The use of rare-earth elements (REEs) has been essential in the technological transition toward a sustainable economy. In various fields, especially, the demands of rare-earth-elements-based permanent magnets, which involve the high consumption of REEs for production, have increased with the advancement of technology [1,2,3,4]. Dysprosium (Dy), which is included in permanent magnets with neodymium (Nd), plays an indispensable role in permanent magnets for preventing the loss of magnetic properties in high-temperature environments [5,6,7]. However, heavy rare-earth elements (HREEs), such as Dy, are more limited resources than light rare-earth elements (LREEs), due to their relatively low composition in ore [8]. These limitations and biases of resources can result in serious shortages in REE supplies. Therefore, the recycling of REEs is rising as an alternative for the achievement of a circular economy.

The recycling processes of REEs are generally divided into hydrometallurgical and pyrometallurgical recycling processes [9]. Even though the high purification of REEs and their recovery efficiency can be obtained by hydrometallurgical recycling, chemical wastes can be generated and lead to serious environmental destruction [10,11,12]. In contrast, the pyrometallurgical process involves the electrical or thermal recovery of REEs as an environmentally friendly alternative without emitting waste, despite a lower recovery efficiency than hydrometallurgy.

In the liquid metal extraction (LME) process, as one of the pyrometallurgical recycling methods, the selective extraction of REEs from *R*FeB magnets (*R* = Nd and Dy) can be induced by the thermodynamically high affinity between REEs and Mg, the extraction agent, compared to other elements, such as Fe and B [13,14,15,16,17]. However, the difference in the extraction efficiency between Nd and Dy is notable. While the extraction of Nd can be completed within 6 h, only 68% of Dy is extracted at 12 h [18,19,20]. The reason for this extractive bottleneck is that Dy–Fe intermetallic compounds and Dy–oxide are generated in the matrix and grain boundary during the reaction between Dy and Mg [18]. In particular, decomposing Dy–Fe intermetallic compounds have been attributed to their dominant role in the initial recovery of Dy, due to the different stabilities of Dy–oxide and Dy–Fe intermetallic compounds [18,19,20]. Even if the control of various process parameters was observed to partially improve the extraction efficiency of Dy, such as adjusting the reaction time, input Mg, and scrap sizes in previous studies [18,19,20], this issue still remains unsettled.

The boro-additive effect (BAE) can serve as a fundamental breakthrough to suppress the generation of Dy–Fe intermetallic compounds by disturbing the reaction between Dy and Fe. B is known as a minor element that is strongly combined with Fe to form Fe_2_B [18,19,20]. It has been indicated that the addition of B disturbs the formation of Dy–Fe intermetallic compounds because of its higher affinity with Fe than any other element in a permanent magnet. However, the pure B metals cannot be decomposed by Mg during the LME process, due to its strong bonding between B atoms. Thus, FeB, iron boride, can be a good catalyst to supply B, instead of pure B metals. Due to its relatively lower stability than Fe_2_B, FeB is decomposed and released with excess B to become Fe_2_B [21,22]. In the LME process, the effect of adding FeB can play a crucial role in releasing excess B in Mg and enhancing the extraction efficiency by suppressing the byproducts.

In the present study, the boro-additive effect (BAE) is thermodynamically described and experimentally verified in ternary and quinary systems. Due to the phase stability, excess B can be produced from the relatively unstable FeB by Mg, and the infiltrated Mg with excess B can be extracted with REEs and the suppression of the Dy–Fe intermetallic phase, which is the extractive hindrance on the LME process. However, it has been found that the growth of the Fe_2_B layer limits the Mg reaction zone and results in the formation of *R*B_4_ as a side effect [23]. By reducing the ratio of FeB, the extraction efficiency of Dy over 12 h with the addition of FeB was improved to 80%, which is almost the same extraction efficiency as the conventional LME process over 24 h.

## 2. Experimental Details

To verify the release of excess B and the BAE by comparing conventional processes, the thermodynamic calculations of ternary and quinary phase diagrams and the sample preparation were carried out at 900 °C. The thermodynamic calculations were conducted by using Factsage 7.1 with FTlite databases and MAGN with private databases. The ternary, quinary, and conventional specimens were prepared at 900 °C for 12 h in a high-frequency induction furnace, as shown in Figure 1a–c, respectively. To control the Dy–Fe intermetallic phase, the optimal temperature and dwelling time at which the Dy–Fe intermetallic compounds started decompose were presented [18,19,20].

Pulverized FeB and magnet scraps were located at the bottom of the crucible, and pure Mg was placed on the FeB and magnet uni-axially to confirm the reaction behavior. The permanent magnets (Jahwa Electronics Co., Ltd., Seoul, Korea) included 25.8 wt.% Nd and 3.5 wt.% Dy as raw materials. Homogeneous FeB (RND Korea), composed of 88.58 wt.% Fe and 16.22 wt.% B, was prepared. Based on a thermodynamic calculation, the ratios of FeB, Mg, and magnet were estimated to be 0.28~0.83, 10, and 1, respectively. The samples were cut in a vertical direction using a wire saw and polished with 400~2000 grid silicon carbide papers and 0.3 micron alumina suspension for sample preparation, to observe the microstructural and phase analysis. For the microstructural and phase analysis, scanning electron microscopy (FE-SEM, Model JSM-100F) with energy dispersive X-ray spectroscopy (EDS) and X-ray diffraction (XRD, Model D/Max 2500PC) were used in point mode, using a 1 mm slit with a Cu target with a wavelength of 1.5406 Å. The complex phase was analyzed by using FULLPROF software with the Le Bail method. The extraction efficiency was estimated by measuring the composition of the REEs in the Mg zone using X-ray fluorescence spectrometry (XRF, Model ARL PERFORM’X).

## 3. Results and Discussion

To confirm the generation of excess B in Mg, the phase stability among Mg, Fe, and B was estimated by thermodynamic calculations. Figure 2 shows the ternary phase diagram for the assessment of the reaction behavior with phase stability at *T* = 900 °C and *P* = 1 atm. This shows that the Fe_2_B phase and Mg boride compounds are the most stable phases, with an adjusted ratio between Mg and FeB among the intermetallic compounds. This indicates that the formation of the Fe_2_B and Mg–B phases is possible by excess B through the mediator Mg, because the FeB phase can be easily decomposed in Mg due to its relatively weak stability.

Based on the possibility of generating excess B from the Mg–FeB reaction on the equilibrium status in a thermodynamic calculation, the experimental model for the ternary phase was investigated in the condition of ratio of 0.83:10 = W_FeB_:W_Mg_, *T* = 900 °C, and *t* = 12 h. Figure 3 shows the microstructure of the results of the reaction between Mg and pulverized FeB to observe the phase transformation. It is divided into three zones, which are colored as light gray (1), dark gray (2), and black (3) in the reacted FeB alloy. The light gray zone (1) is the region in which it is possible to mainly distribute Fe_2_B as the interfacial zone between Mg and FeB, the dark gray zone (2) is the remaining region that cannot react as much with Mg as the original bulk FeB, and the black zone (3) is the infiltrated trace of Mg through the cracks caused by the initial FeB sample preparation. While the composition in zone (2) is consistent with the initial FeB composition in Appendix A, the Fe contents in zone (1) are definitely increased to 90.66 wt.% from 82.66 wt.% in Table 1. This result is in good agreement with the converted value of the atomic weight of Fe_2_B with Fe:B = 65.58:34.42. The phase analysis in zone (1) and zone (2) is exhibited in Figure 4a,b, respectively. Even though the XRD measurement cannot be perfectly separated between zones (1) and (2) due to the limit of the slit size, it is inferred that the Fe_2_B phase is dominant in zone (1) rather than (2), by the difference in the intensity of Fe_2_B. The peaks of Fe_2_B, FeB, and Mg are specified as blue triangles, black squares, and red circles in Figure 4. The original peaks of Fe_2_B and FeB mainly appear in Figure 4a,b, respectively, even if some peaks overlap with each other. Figure 5 shows the magnified microstructure of zone (3) with colored EDS mapping to distinguish components, which are B, Mg, and Fe, in green, red, and sky blue, respectively. The cracks caused by the pulverized FeB definitely offer the diffused route for Mg. However, the B contents and Mg contents are dominant in zone (3), as shown in Table 1. Thermodynamically, the excess B from the FeB can be diffused to the liquid Mg as an intermediate state, because MgB_2_ exhibits a stable phase in the equilibrium state. It is indicated that the liquid state of Mg leads not only to the depletion of B in the relatively unstable FeB phase, but also to the conveying role.

To verify the B additive effect based on the release of excess B, thermodynamic calculations on the quinary system are conducted to simulate the reaction by FeB addition under conventional conditions of 1:10 (W_Magnet_:W_Mg_), as shown in Figure 6. The quinary phase diagram shows that the phase of *R*–Fe intermetallic compounds can be suppressed by BAE, only in the range of 0.0683 < FeB < 0.0720. The adequate range of FeB addition can be supported to release excess B to Mg, and the reaction between REEs and Fe is disturbed due to the high reactivity between Fe and B in the LME process.

The LME experiment of the quinary system is designed based on the conventional conditions of *T* = 900 °C, *t* = 12 h, and 1:0.83:10 (W_Magnet_:W_FeB_:W_Mg_), which is approximately 0.07 of the FeB composition in the phase diagram. Generally, in the conventional LME process, the magnet scrap keeps its shape as an input raw material with Mg ligaments after the reaction, because REEs selectively diffuse into liquid Mg, and Fe remains in Figure 7a [13,14,15,16,17,18,19,20]. However, the microstructure of the magnet scraps with the addition of FeB shows two zones, which are composed of the Fe–B zone and the reacted zone in the magnet scraps in Figure 7b. To analyze those phases, Figure 8 and Table 2 show the magnified microstructure and the investigated composition in each zone. Figure 8b shows that the scattered particles in the reacted zone are simply characterized as the Fe_2_B phase. Mg, including excess B, can infiltrate the magnet, and Fe_2_B can be generated because the remaining Fe in the magnet is combined with B before forming Dy–Fe bonds. Thus, this suggests that the Dy–Fe intermetallic phase, which is the reason for the extractive bottleneck, can be controlled by the supported BAE on the conventional LME process. On the other hand, there is a decisive shortcoming, which is the phase of the Fe–B zone in Figure 8c. The characterized composition in the Fe–B zone is quite different from that in the reacted zone in Table 2. The complex phases are analyzed by the refinement of the XRD pattern in Figure 9. While the reacted zone is composed of only Fe_2_B and Mg, as shown in Figure 9a, the phase analysis of the Fe–B zone shows Fe_2_B phases, and REEs–B intermetallic compounds are detected as *R*B_4_ phases, as shown in Figure 9b. Interestingly, even though REEs–B intermetallic compounds cannot be fundamentally formed at 900 °C due to the high affinity between Fe and B, as shown in Figure 6, Fe_2_B undergoes a growth of the boride layer at 900 °C because of the abundant FeB environment during the phase transformation from FeB to Fe_2_B [21,22]. The REE extraction efficiency is inevitably suppressed because the growth of the layered Fe_2_B results in the formation of *R*B_4_, which is confirmed as the most stable phase in the *R*–B system [23] and blocks Mg infiltration. Thus, FeB not only plays a crucial role in removing extractive hindrance, but also blocks the diffusion route of Mg by simultaneously producing other byproducts.

Even though there is compensation for REEs–Fe intermetallic compounds, the amount of FeB is reduced to avoid the growth of the Fe_2_B layer in Appendix A. To observe the distribution of the intermetallic phases inside the magnet, the microstructures of the specimen of the conventional LME process and the specimen of the optimized FeB ratio are compared in Figure 10a,b, respectively. The Dy–Fe intermetallic components in the matrix of magnets are predominant in the microstructure of the conventional process in Table 3 [18,19,20]. On the other hand, in the case of the FeB addition process, the magnet turned to Fe_2_B with partially removed Dy–Fe intermetallic compounds in the reacted zone, due to the reduction of the FeB ratio. Thus, it is found that the Dy contents significantly decreased by 21.42 wt.% using the supported BAE in Table 3.

To observe the catalytic effect of the BAE on the extraction efficiency of REEs by removing Dy–Fe intermetallic compounds, the extraction efficiency was calculated with Equations (1)–(4). The extraction efficiency of REEs, which includes all the REEs in the magnet, was calculated by Equations (1) and (2), and the extraction efficiency of Dy was calculated individually by Equations (3) and (4). By calculating the ratio between the total weight of the REEs (W_REEs_) and the total weight of the Mg–REEs alloy (W_Mg_ + W_REEs_), the composition of Mg when the REEs were totally extracted (C_100%_) can be calculated using Equation (1). The extraction efficiency of the REEs can be estimated as the ratio between the composition of REEs in Mg (C_REEs_) and C_100%_, as shown in Equation (2). C_REEs_ is the XRF result of the Mg zone after the reaction experiment. The calculation of the extraction efficiency of Dy is equally estimated as follows:(1)$$C_{100\%\;\mathrm{of}\;\mathrm{REEs}}\;\left(\mathrm{wt}.\%\right)=\left(\frac{W_{\mathrm{REEs}\;\mathrm{in}\;\mathrm{Magnet}}}{W_{\mathrm{REEs}\;\mathrm{in}\;\mathrm{Magnet}}{+W}_{\mathrm{Mg}}}\;\times \;100\right)$$
(2)$$\mathrm{Extraction}\;\mathrm{Efficiency}\;\mathrm{of}\;\mathrm{REEs}\;\left(\%\right)=\left(\frac{C_{\mathrm{REEs}\;\mathrm{in}\;\mathrm{Mg}}}{C_{100\%\;\mathrm{of}\;\mathrm{REEs}}}\;\times \;100\right)$$
(3)$$C_{100\%\;\mathrm{of}\;\mathrm{Dy}}\;\left(\mathrm{wt}.\%\right)=\left(\frac{W_{\mathrm{Dy}\;\mathrm{in}\;\mathrm{Magnet}}}{W_{\mathrm{Dy}\;\mathrm{in}\;\mathrm{Magnet}}{\;+\;W}_{\mathrm{Mg}}}\;\times \;100\right)$$
(4)$$\mathrm{Extraction}\;\mathrm{Efficiency}\;\mathrm{of}\;\mathrm{Dy}\;\left(\%\right)=\left(\frac{C_{\mathrm{Dy}\;\mathrm{in}\;\mathrm{Mg}}}{C_{100\%\;\mathrm{of}\;\mathrm{Dy}}}\;\times \;100\right)$$

C_100% of REEs_, C_100% of Dy_, C_REEs in Mg_, C_Dy in Mg_, and the extraction efficiency are indicated in Table 4. The extraction efficiency of Dy was approximately 17% higher in the FeB addition process than in the conventional process. In addition, the extraction efficiency of Dy in the FeB addition process over 12 h was similar to that of the conventional LME process over 24 h. It is suggested that the LME process with an FeB addition can be a decisive parameter in improving the energy consumption for processing and enhancing the extraction efficiency of REEs.

## 4. Conclusions

The extraction of Dy using the conventional LME process results in a bottleneck, due to formation of Dy–Fe intermetallic compounds by an affinity between Dy and Fe. The disturbance of the affinity between Dy and Fe is necessary to improve Dy extraction by eliminating Dy–Fe intermetallic compounds. In order to suppress the formation of Dy–Fe intermetallic compounds, a highly reactive element with Fe, rather than Dy, is needed, and B has the highest affinity with Fe in the systems. Therefore, the boro-additive effect (BAE) on the LME process is thermodynamically designed and examined in ternary and quinary systems in this paper. The evaluation of the thermodynamic approaches in a ternary system (Mg–Fe–B) describes the decomposition behavior of FeB, which is the production of excess B by phase stability. It was observed that the depletion of B in FeB and excess B can be infiltrated into magnets with Mg, due to the relatively unstable FeB phase. Based on the principle of excess B and the thermodynamic prediction of the quinary system in the LME extraction process, it was revealed that the formation of Fe_2_B can affect the extraction efficiency of Dy by disturbing Dy–Fe bonding. However, the growth of layered Fe_2_B from the phase evolution of FeB and the magnet resulted in the downsizing of the Mg region, which mainly reacted with RE from the magnet. The loss of extraction resulted in the formation of *R*B_4_ between the REEs in the magnets and excess B in the isolated Mg puddles. The reduced ratio of FeB to one-third effectively conveyed excess B inside the magnet, forming Fe–B intermetallic compounds without any side effects. Although the Dy–Fe intermetallic components cannot be totally controlled, it was observed that the generation of *R*B_4_ from the growth of the Fe_2_B layer in the reacted zone was suppressed. As a result, the extraction efficiency of Dy was approximately 17% higher in the FeB addition process than in the conventional process, and 12 h can be saved, compared to the conventional method.

## Figures and Tables

**Figure 1 materials-15-03032-f001:**
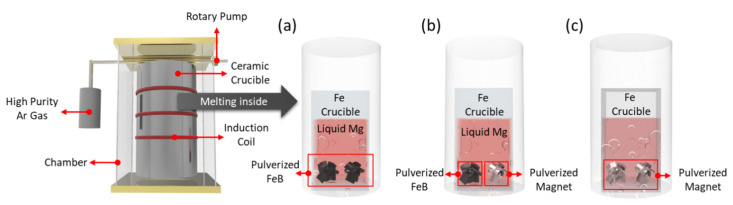
Schematics of the reaction experiments of (**a**) Mg–FeB, (**b**) Mg–FeB–magnet, and (**c**) conventional LME process.

**Figure 2 materials-15-03032-f002:**
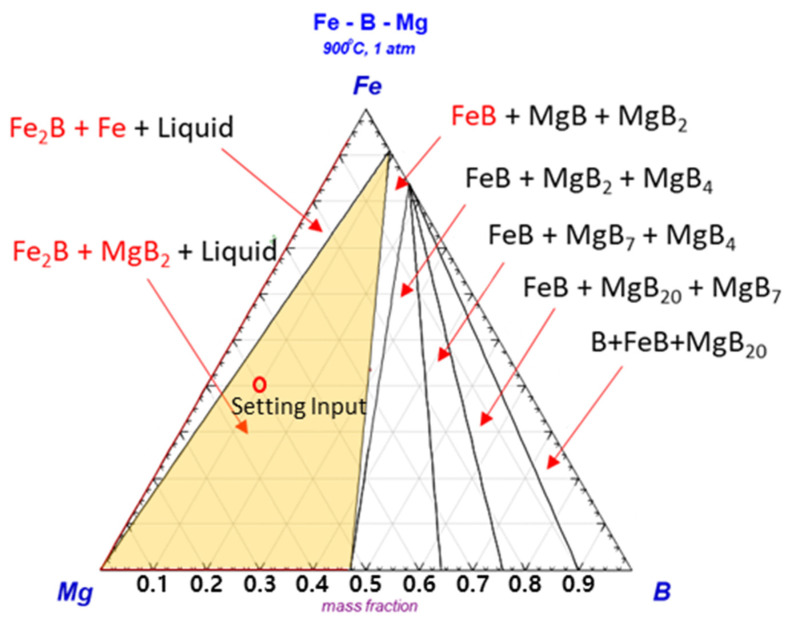
Ternary phase diagram of Mg–Fe–B at 900 °C under 1 atm by FactSage.

**Figure 3 materials-15-03032-f003:**
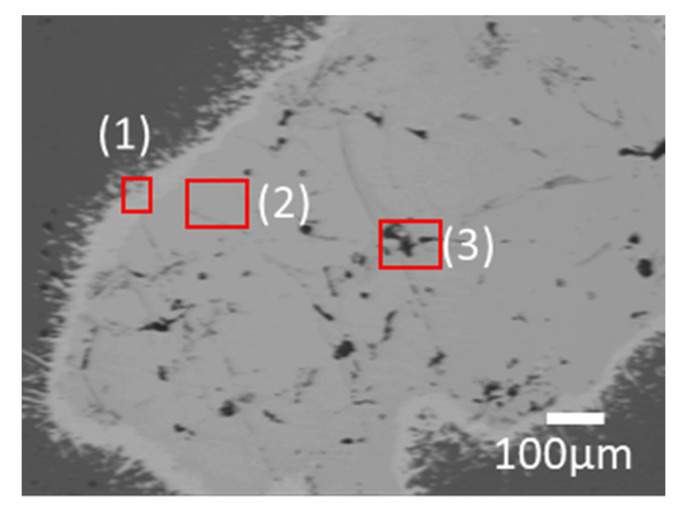
Microstructure of FeB after the reaction experiment between Mg and FeB (×50 magnification).

**Figure 4 materials-15-03032-f004:**
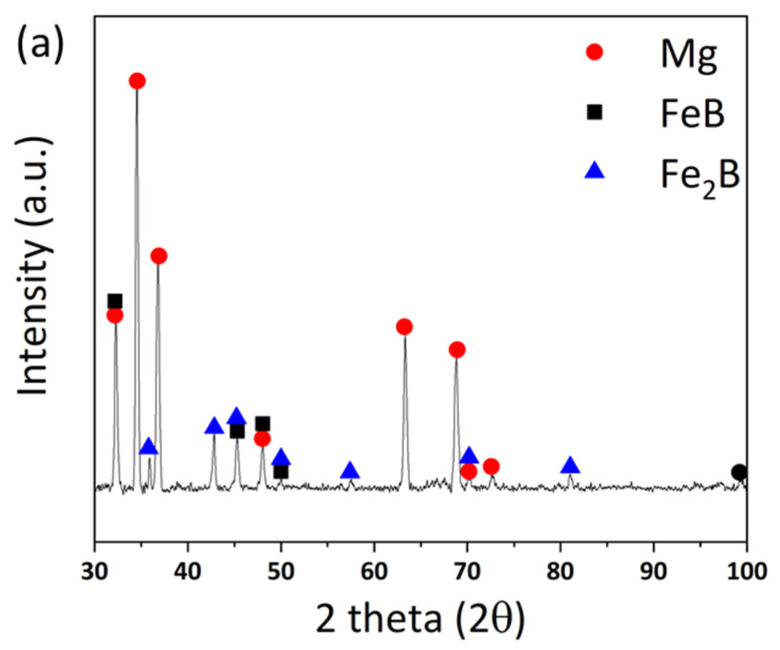
Phase analysis with point XRD in (**a**) light gray phase-dominated zone, as (1) in Figure 3, and (**b**) dark gray phase-dominated zone, as (2) in Figure 3.

**Figure 5 materials-15-03032-f005:**
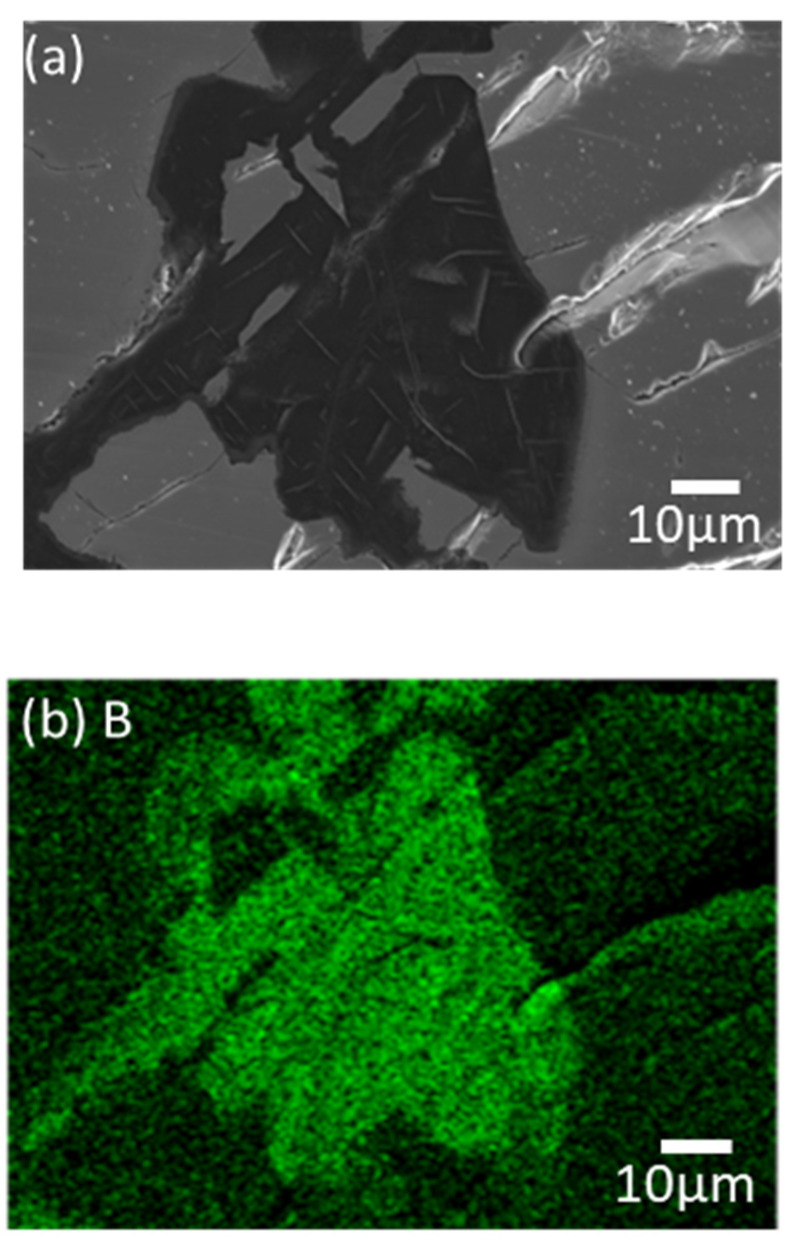
(**a**) Enlarged microstructure of (3) in Figure 3 and EDS mapping results of (**b**) B, (**c**) Mg, and (**d**) Fe (×500 magnification).

**Figure 6 materials-15-03032-f006:**
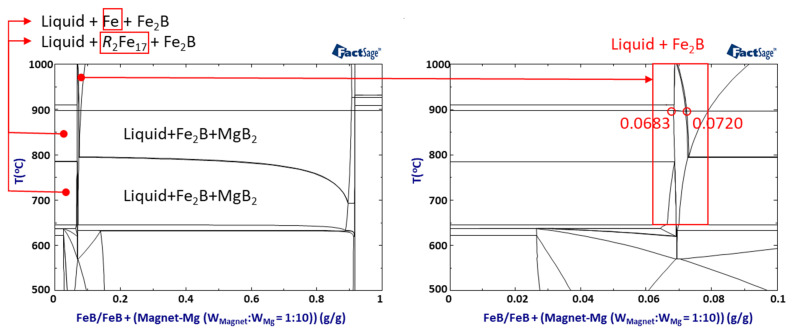
Thermodynamic calculation of Mg–FeB–magnet with an optimized weight ratio between the magnet and Mg as 1:10 (W_Magnet_:W_Mg_).

**Figure 7 materials-15-03032-f007:**
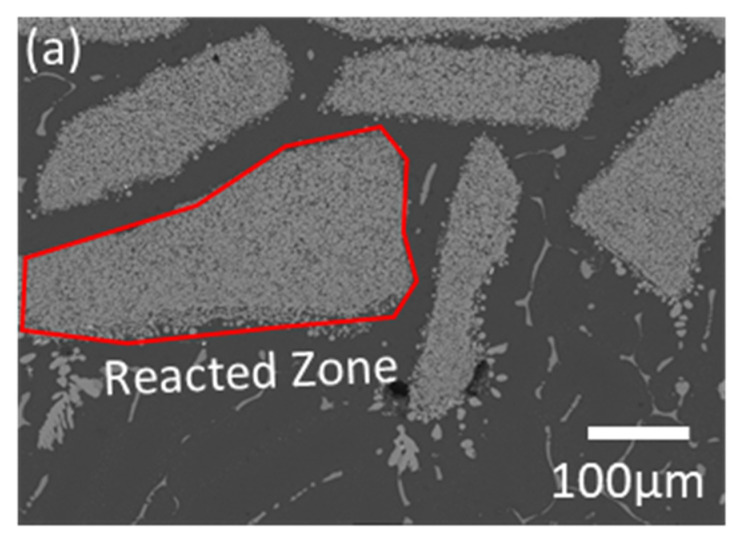
Microstructures after the reaction experiment with (**a**) a conventional LME process and (**b**) FeB-added LME process (×100 magnification).

**Figure 8 materials-15-03032-f008:**
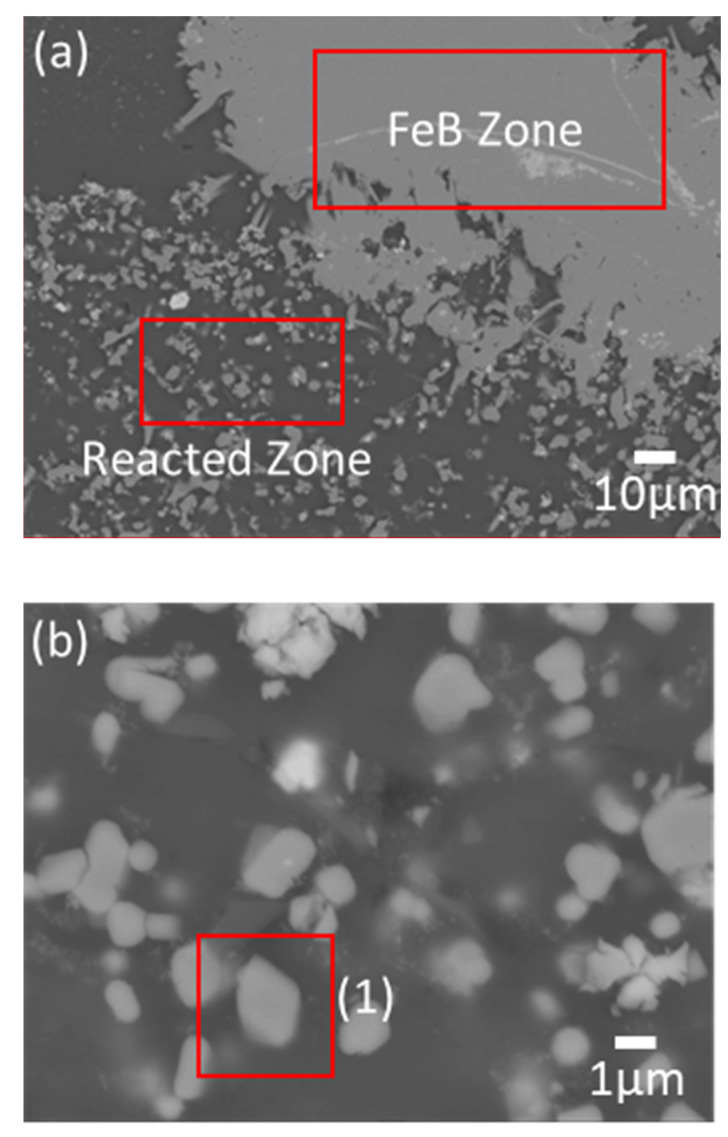
Overall microstructures of the (**a**) reacted zone (×500 magnification), FeB zone, and enlarged microstructures of the (**b**) reacted zone and (**c**) FeB zone (×5000 magnification).

**Figure 9 materials-15-03032-f009:**
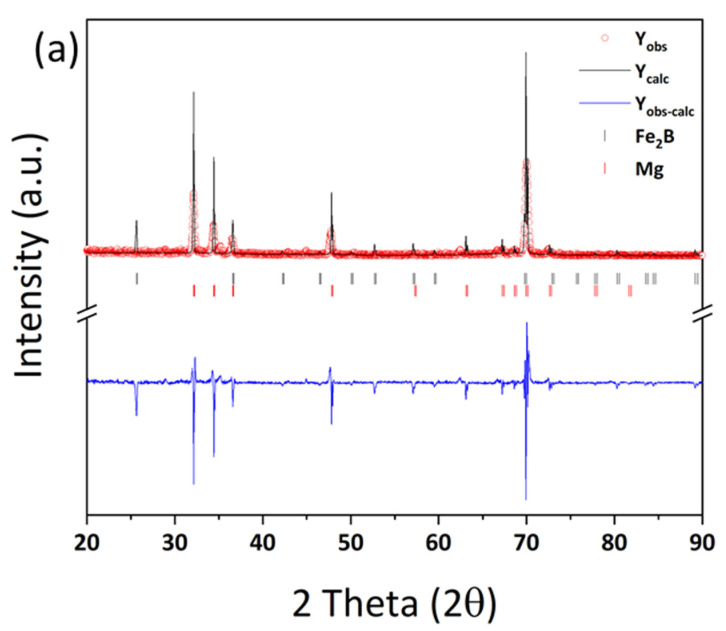
Phase analysis of (**a**) the reacted zone, as (1) in Figure 8, and (**b**) FeB zone, as (2) in Figure 8, using point XRD with Le Bail refinement by FULLPROF.

**Figure 10 materials-15-03032-f010:**
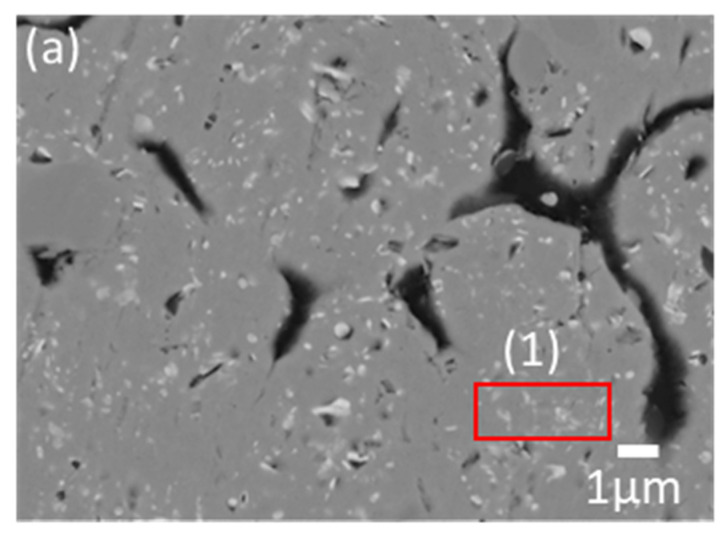
Comparison between the microstructures inside the magnet from the (**a**) conventional LME process and (**b**) FeB-added LME process (×5000 magnification).

**Table 1 materials-15-03032-t001:** EDS results for observing the composition of phases in (1), (2), and (3) of Figure 3.

Composition (wt.%)	Mg	Fe	B	Minor Elements
(1)	-	90.66	9.21	0.13
(2)	-	82.99	16.92	0.09
(3)	19.37	-	80.56	0.07

**Table 2 materials-15-03032-t002:** EDS results for observing the composition of phases in (1) of Figure 8b and (2) of Figure 8c.

Composition (wt.%)	Nd	Dy	Fe	B	Minor
(1)	-	-	92.25	7.52	0.23
(2)	64.48	3.30	-	31.11	1.11

**Table 3 materials-15-03032-t003:** EDS results for observing the composition of phases in (1) of Figure 10a and (2) of Figure 10b.

Composition (wt.%)	Nd	Dy	Fe	B	Minor
(1)	-	22.68	75.39	-	1.93
(2)	-	1.26	83.91	13.61	1.22

**Table 4 materials-15-03032-t004:** Concentration and extraction efficiency of REEs and Dy in the conventional process and FeB addition process.

	Conventional Process	FeB Addition
C_100% of REEs_	2.847 wt.%	
C_100% of Dy_	0.349 wt.%	
C_RE in Mg_	2.753 wt.%	2.795 wt.%
C_Dy in Mg_	0.238 wt.%	0.279 wt.%
Extraction efficiency of REEs	96.70%	98.17%
Extraction efficiency of Dy	68.24%	80.02%

## Data Availability

Not applicable.

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
