# Peer review of "The Supported Boro-Additive Effect for the Selective Recovery of Dy Elements from Rare-Earth-Elements-Based Magnets"

_materials, 2022, doi:10.3390/ma15093032_

Round 1

Reviewer 1 Report

Review

Manuscript ID:  materials-1680180

Journal: materials

Title:

 The supported boro-additive effect for the selective recovery of Dy elements from rare earth-based magnets

The author try to study  Liquid metal extraction (LME) for recycling rare earth elements from magnets has been studied to examine its suitability as an environmentally friendly alternative for a circular economy. While Nd(neodymium) extraction efficiency can easily reach almost 100% based on the high reac-tivity of Mg(magnesium), Dy(dysprosium) extraction has been limited because of the Dy-Fe inter-metallic phase as the main extractive bottleneck. Herein, the boro-additive effect is designed ther-modynamically and examined in the ternary and quinary systems to improve the selectivity of re-covery. Based on the strong chemical affinity between B (boron) and Fe, the effect of excess boron, which is produced by the depletion of B in FeB by Mg, successfully resulted in the formation of Fe2B instead of Dy-Fe bonding. But there are some comments as the following:

  • The author must modify figures 1,2 due to the resolution is very bad
  • The authors must modify figure 3 due to absence of magnification power
  • The authors must modify figure 5 due to absence of magnification power
  • The authors must modify tables 1-4 due to the resolution is very bad
  • the resolution must add more characterization such as XPS

Recommendation: Minor revision

Author Response

Reviewer 1

Thanks for reviewing this paper with efforts. These comments are very helpful for editing this paper in detail. 
The authors reflect the comments with editing more detail contents in paper.

Comment 1 and 4. Bad Resolution
We are sorry to say that resolution of figures and tables were downgraded in conversion process from pdf file to doc file. 
Therefore, we put whole raw figures and tables to doc file directly as shown in paper.
Also, axis of ternary phase diagram was enlarged for observing easily.

Comment 2 and 3. Magnification
Thanks for the helpful comment to make detail explanation.
We described whole magnification of figures including Fig 3 and 5.

Comment 5. Additional Characterization
XPS is considered as very powerful analysis for characterization of atomic bonding.
We are very thankful for your recommendation but XPS was hard to be applied with the reason that phases in our samples are not homogeneous and also consist micro-scaled system.
Boro-additive effect is mainly examined by observing the behavior of the phase transformation and microstructural studies during LME process.
The transformed phases were characterized by XRD analysis using Fullprof software which is advanced analytics to confirm phases.
Evaluations for recovery efficiency and quantative analysis were observed SEM-EDS and XRF.
In order to improve resolution, SEM-EDS and XRF were tried over 10 times by region and error range was about 0.5wt.%.
In addition, our selected advanced analytics followed Okabe et al, Akahori et al [14-16] who are trailblazers in pyro-metallurgical field.
Okabe et al. and Akahori et al. also showed recovery efficiency with XRF and quantative analysis with SEM-EDS. In addition, we added XRD for improving resolution with Fullprof.
Therefore, we considered that these analysis were adequate to describe our results.
There is a limitation as we mentioned right now however, we are trying to apply various analytics as you mentioned.
Thank you for recommendation

Reviewer 2 Report

The manuscript by Park et al. investigates the boro-additive effect (BAE) which is thermodynamically described and experimentally verified in ternary and quinary systems. The evaluation of thermodynamic approaches in a ternary system (Mg-Fe-B) describes the decomposition behavior of FeB, which is the production of excess B by phase stability. The authors observe the depletion of B in the FeB and the excess B can be infiltrated into magnets with Mg due to the relatively unstable FeB phase. The excess B principle and thermodynamic prediction of the quinary system reveals that the formation of Fe2B can affect the extraction efficiency of Dy by disturbing Dy-Fe bonding. The authors also observe the growth of the Fe2B layer limited by the Mg reaction zone whih gives rise to the formation of RB4 as a side effect. The generation of RB4 from the growth of the Fe2B layer in the reacted zone is suppressed. By reducing the ratio of FeB, the extraction efficiency of Dy over 12 hours with FeB addition was improved to 80%, which is almost the same extraction efficiency as the conventional LME process over 24 hours.

Overall, this is contemporary research in the context of LME for recycling rare earth elements from magnets. The paper is organized and well supported by the results, adequate characterization, and conclusions. I also appreciate the scholarly interpretation of the authors in the analysis of all the electron microscopy data and microstructure analysis throughout the main text. I recommend the manuscript for publication after addressing the comments listed below.

Major

  1. The XRD data in fig. 5a-b are poorly interpreted. For e.g. – What does the symbols for respective peak indicate? Please use different colors and specify the peaks and phases in the figures?

  1. The authors should briefly describe the scope and challenges for this work in the conclusion section.

  1. The authors should include a methods section describing the instrumentation and sample preparation for SEM and XRD experiments?

Author Response

Reviewer 2

Thanks for reviewing this paper with efforts. These comments are very helpful for editing this paper in detail. 
The authors reflect the comments with editing more detail contents in paper.
In addition, we would like to summarize edited comments as below.

Comment 1. The XRD data in fig. 5a-b are poorly interpreted. For e.g. – What does the symbols for respective peak indicate? Please use different colors and specify the peaks and phases in the figures?

In order to specify peaks in XRD, we changed colors, shapes and sizes of indication in Figure 4. In addition, we explained these changes in detail.

Comment 2. The authors should briefly describe the scope and challenges for this work in the conclusion section.

Ultimate goal for this paper was disturbance of affinity between Fe and Dy using highly reactive element with Fe than Dy to improve Dy extraction during LME process. Therefore, we select B as breakthrough for elimination Dy-Fe intermetallic compounds and prove boro-additive effect with thermodynamic and practical assessments.
we put contents about these explanation more in detail in conclusion section.

Comment 3. The authors should include a methods section describing the instrumentation and sample preparation for SEM and XRD experiments?

We didn't introduce sample preparation even if explanation about SEM and XRD was described. Sample was prepared with polishing processes and we described sample preparation in detail.